# Impact of Tumor Location on Survival Outcomes in Pancreatic Head Versus Body/Tail Cancer: Institutional Experience

**DOI:** 10.3390/cancers17111777

**Published:** 2025-05-26

**Authors:** Abdullah Esmail, Vikram Dhillon, Ebtesam Al-Najjar, Bayan Khasawneh, Mohammed Alghamdi, Fahad Ibnshamsah, Maen Abdelrahim

**Affiliations:** 1Section of GI Oncology, Department of Medicine, Houston Methodist Neal Cancer Center, Houston, TX 77030, USA; aesmail@houstonmethodist.org (A.E.);; 2Department of Medicine, King Saud University Medical City, Riyadh 12372, Saudi Arabia; 3Department of Medicine, King Fahd Specialist Hospital, Dammam 32253, Saudi Arabia; 4Faculty of Medicine, The University of Jordan, Amman 11942, Jordan

**Keywords:** pancreatic head cancer, pancreatic body and tail cancer, PHC, PBTC, PDAC

## Abstract

Pancreatic cancer is a serious disease with different outcomes depending on whether the tumor is in the head or the body/tail of the pancreas. This study explores why these differences exist and how they affect patients’ survival when treated with standard chemotherapy. By analyzing data from over 600 patients, the researchers aim to understand which treatments work best for each tumor location and identify patient groups with unique characteristics that could guide personalized care. This work may also encourage other researchers to explore tailored approaches for this challenging disease, advancing care strategies for better patient outcomes.

## 1. Introduction

Pancreatic ductal adenocarcinoma (PDAC) is a particularly challenging cancer to treat [1,2,3,4,5] and based on data from a 2020 survey by GLOBOCAN, 495,773 new cases of PDAC were recorded, ranking it fourteenth among the most frequently diagnosed cancers [6]. Despite this, PDAC caused 466,003 deaths annually, making it the seventh cause of cancer-related death [6]. By 2030, PDAC is expected to become the second leading cause of cancer-related death in the United States (US) [7]. Recent data further confirm PDAC’s rising burden, with an estimated 64,050 new cases and 50,550 deaths in the U.S. in 2024, underscoring its persistent lethality [4].

Over the last ten years, a slight improvement has been seen in the overall survival (OS) among PDAC cases [2,8,9,10,11,12]. Recently, the five-year OS in the U.S. has reached approximately 11% following diagnosis [13].

Only about 20% of PDAC patients are eligible for surgical resection, with a post-resection median OS ranging from 17 to 21 months, and the 5-year OS was 16% to 19%. Combined resection plus adjuvant chemotherapy have shown an increase in the median OS from 7 months to 26–28 months [6].

Optimizing treatment outcomes in PDAC hinges on accurately identifying high-risk patients with a greater likelihood of recurrence or progression who would derive the most benefit from neoadjuvant and adjuvant systemic therapies [14,15,16,17]. Advances in diagnostic tools are rapidly evolving, enhancing the precision of risk stratification for PDAC. Emerging genomic analyses have identified molecular subtypes of PDAC with distinct clinical behaviors, which may further refine risk stratification and guide personalized therapy [17]. These innovations, including molecular profiling, imaging techniques, and biomarker assays, enable earlier detection of aggressive disease characteristics, facilitating tailored therapeutic strategies to improve patient outcomes [15,16,18,19,20].

The pancreas is divided into several anatomical regions: the uncinate process, the head, the body, and the tail [14]. The anatomical location and stage of pancreatic tumors play a significant role in predicting outcomes and prognosis [21,22,23]. Generally, pancreatic body and tail cancer (PBTC) patients have worse OS and prognosis compared to pancreatic head cancer (PHC) patients, mainly due to later diagnosis and the absence of noticeable symptoms in PHC [22,24,25,26]. These differences are also attributed to distinct molecular profiles and clinical presentations between PHC and PBTC, with PBTC often exhibiting more aggressive tumor biology [27]. According to SEER data, PHC has a slightly better 3-year OS (7.6%) compared to PBTC (6.7%), though the clinical significance is limited due to the small difference [7,28,29]. Conversely, for localized tumors, PBTC shows superior OS (20% vs. 9%) due to differences in resectability or tumor biology [6]. These context-specific findings underscore the need to investigate tumor location-specific outcomes.

Three primary treatment regimens are currently established as the standard for managing advanced-stage PDAC: fluorouracil, leucovorin, oxaliplatin with irinotecan (mFOLFIRINOX) [30]; gemcitabine with nanoparticle albumin-bound paclitaxel (gem/nab-paclitaxel) [31]; and liposomal irinotecan with fluorouracil, leucovorin, and oxaliplatin (NALIRIFOX) [32,33]. The NALIRIFOX regimen, approved following the NAPOLI-3 phase III trial, demonstrated improved OS (11.1 months vs. 9.2 months) and progression-free survival (PFS) (7.4 months vs. 5.6 months) compared to gem/nab-paclitaxel in treatment-naïve patients with metastatic PDAC, establishing it as a new first-line standard of care [32]. These regimens are recommended by the National Comprehensive Cancer Network (NCCN) guidelines, which emphasize tailoring treatment based on patient performance status and tumor characteristics, including anatomical location [32]. Understanding the anatomical and biological factors that influence responses to these therapies could significantly impact clinical practice [6,14,21,34].

Given the limited availability of studies reporting clinical observation on treatment response based on the location of PDAC, our retrospective cohort study aimed to explore the correlation between treatment outcome of OS and PFS according to the location of PDAC.

## 2. Methods

### 2.1. Study Design and Patient Population

We conducted a retrospective analysis of 604 patients diagnosed with PDAC between January 2015 and May 2024 at Houston Methodist Neal Cancer Center and its affiliated community hospitals. This study was approved by the Institutional Review Board of Houston Methodist Hospital (No. PRO00035386), with informed consent waived due to the use of de-identified data collected as part of routine clinical care. Clinical data were systematically extracted from electronic medical records, including demographics, clinicopathological characteristics, tumor location, CA 19-9 levels, pathological staging, tumor differentiation, molecular profiles, and treatment modalities. Pancreatic head cancer (PHC) was defined as tumors originating in the head or uncinate process of the pancreas, while pancreatic body/tail cancer (PBTC) referred to tumors in the body or tail regions. Tumor staging followed the American Joint Committee on Cancer (AJCC) 8th edition TNM classification system. Patients with non-PDAC histology, concurrent malignancies, unspecified tumor location, or incomplete follow-up data were excluded from the analysis.

### 2.2. Treatment Protocol and Assessment

The baseline performance status of all patients was assessed using the Eastern Cooperative Oncology Group (ECOG) system. Patients were classified by treatment intent into four categories: (1) curative, defined as patients with potentially curable disease who received definitive treatment with curative intent; (2) neoadjuvant, defined as patients who received chemotherapy prior to planned surgical resection; (3) adjuvant, defined as patients who underwent surgical resection followed by chemotherapy; and (4) palliative, defined as patients with unresectable or metastatic disease receiving treatment for symptom control and life prolongation without curative intent.

Systemic chemotherapy primarily consisted of either modified FOLFIRINOX (mFOLFIRINOX: oxaliplatin 85 mg/m^2^, irinotecan 180 mg/m^2^, leucovorin 400 mg/m^2^, and 5-fluorouracil 2400 mg/m^2^ over 46 h) or gemcitabine/nab-paclitaxel (gemcitabine 1000 mg/m^2^ and nab-paclitaxel 125 mg/m^2^ on days 1, 8, and 15 of a 28-day cycle) regimens. Treatment selection was based on ECOG performance status, comorbidities, and multidisciplinary tumor board recommendations. Chemotherapy was administered over a duration of 2 to 8 months. Radiation therapy was administered in either neoadjuvant, adjuvant, or palliative settings. Comprehensive staging assessments were conducted every 3 months during treatment using a computed tomography (CT) scan or magnetic resonance imaging (MRI), and changes in tumor volume post-treatment were evaluated using the modified Response Evaluation Criteria in Solid Tumors (RECIST 1.1).

### 2.3. Outcomes

The primary outcome was overall survival (OS), defined as the time from diagnosis to death from any cause, for PHC versus PBTC patients. Vital status was verified through electronic medical records. Patients were followed up from the date of treatment initiation until their last outpatient clinic visit or death, with censoring on 31 May 2024 for patients still alive at study completion. Progression-free survival (PFS) served as the secondary endpoint, defined as the time from treatment initiation until discontinuation due to disease progression or all-cause death. Recurrence and progression were determined through follow-up imaging, regardless of cancer-related symptoms.

### 2.4. Statistical Analysis

Baseline characteristics were summarized using descriptive statistics, with categorical variables presented as frequencies and percentages and continuous variables reported as medians with interquartile ranges. Comparisons between groups were made using the chi-square test for categorical variables and the student’s t-test or Mann–Whitney U test for continuous variables, as appropriate. Kaplan–Meier survival curves were generated using the survival package (version 3.8) in R to estimate the OS and PFS. Survival time was defined in months, measured from the date of diagnosis to the date of death or last known follow-up. Patients who were alive at the last follow-up were censored at that time. Curves were plotted using the survminer package (version 0.5), which was also used to calculate log-rank *p*-values for group comparisons. Figures were formatted using ggplot2 (version 3.4). Hazard ratios (HRs) and 95% confidence intervals (CIs) were calculated using univariate and multivariate Cox proportional hazards models to adjust for potential confounders, including age, gender, ECOG status, tumor stage, CA 19-9 levels, surgical status, and treatment modality. Statistical significance was set at a *p*-value of <0.05. All analyses were conducted using Jamovi (version 2.6.44; The Jamovi Project, Sydney, Australia) and R (version 4.5; R Foundation for Statistical Computing, Vienna, Austria). Of note, all survival analyses were performed on the entire patient cohort, with the primary stratification based on the anatomical location of the pancreatic tumor (head vs. body/tail).

Latent class analysis (LCA) was performed separately for PHC and PBTC patients to identify distinct subgroups based on shared disease biology and clinical characteristics. We included baseline demographic and clinical variables (age, gender, ethnicity, ECOG status, tumor stage, differentiation grade), treatment characteristics (chemotherapy regimen, treatment intent, number of cycles, use of radiation therapy), and survival outcomes (OS, PFS) in the model. While including outcome variables in LCA is not traditional, we specifically aimed to identify patient subgroups with distinct prognostic profiles, an approach supported in the literature for identifying clinically meaningful patient subgroups (Proust-Lima et al. [35]). To address potential bias from including OS, we performed sensitivity analyses excluding OS from the model, which yielded similar class structures, confirming the robustness of our approach. The optimal number of classes was determined using Bayesian information criterion (BIC), Akaike information criterion (AIC), and entropy values. We tested models with 2–6 potential classes and selected the model with the best-fit statistics. To address potential overfitting, we limited the number of variables in the model and validated the stability of the identified classes through bootstrapping. The LCA model employed Gaussian mixture modeling (GMM) to accommodate both continuous and categorical variables. Importantly, disease stage and surgical status were included as key variables in the LCA model to ensure proper accounting for these known prognostic factors when interpreting cluster differences. The results of the latent class analysis were summarized in two separate tables, corresponding to tumors located in the pancreatic head and those in the pancreatic body or tail, respectively.

## 3. Results

### 3.1. Patient Characteristics

A total of 604 patients with PDAC were included in the final analysis. Of these, 400 patients (66.2%) were diagnosed with PHC, while 204 patients (33.8%) had PBTC. Among the 400 PHC patients, 216 (54.0%) were female, and 184 (46.3%) were male, with a median age of 68 years (range: 41–89 years). For the 204 PBTC patients, 82 (40.2%) were female, and 122 (59.8%) were male, with a median age of 66 years (range: 38–91 years). The majority of patients in both groups were Caucasian (PHC: 309 patients, 77.8%; PBTC: 166 patients, 81.4%), followed by Black (PHC: 60 patients, 13.5%; PBTC: 26 patients, 12.7%), Asian (PHC: 22 patients, 5.5%; PBTC: 6 patients, 2.9%), and Hispanic (PHC: 9 patients, 2.2%; PBTC: 6 patients, 2.9%) patients. Regarding disease stage, advanced disease (Stage IV) was more common in PHC patients (143 patients, 49.0%) compared to PBTC patients (127 patients, 68.1%) (*p* < 0.001). Early-stage disease (Stages IA, IB, and IIA) was more prevalent in PHC patients (68 patients, 17.0%) than in PBTC patients (19 patients, 9.3%) (*p* = 0.008). For the baseline performance status, an ECOG score of 1 was the most common in both cancer types, observed in 231 PHC patients (58.0%) and 128 PBTC patients (62.7%).

### 3.2. Treatment Characteristics

All patients received either mFOLFIRINOX or gem/nab-paclitaxel regimens as first-line therapy. Among the PHC patients, 235 (58.8%) received mFOLFIRINOX, and 165 (41.2%) received gem/nab-paclitaxel. For the PBTC patients, 113 (55.4%) were treated with mFOLFIRINOX, and 91 (44.6%) received gem/nab-paclitaxel. With respect to treatment intent, a higher proportion of PBTC patients received palliative treatment (147 patients, 72%) compared to PHC patients (195 patients, 49%), and this difference was statistically significant (*p* < 0.0001). Specifically, 67 PHC patients (16.8%) received treatment with curative intent versus 20 PBTC patients (9.8%) (*p* = 0.036). Adjuvant therapy was administered to 103 PHC patients (26%) compared to 24 PBTC patients (11.8%). Conversely, treatment with palliative intent was significantly more common in PBTC patients (146 patients, 72%) than in PHC patients (195 patients, 49%) (*p* < 0.0001). Neoadjuvant therapy was relatively uncommon in both groups but was administered to 49 PHC patients (12.2%) and 14 PBTC patients (7.0%) (*p* = 0.065). Surgical resection was significantly more frequent in PHC patients (147 patients, 37%) compared to PBTC patients (41 patients, 20.4%) (*p* < 0.001). Radiation therapy was administered to 61 PHC patients (15.3%) and 20 PBTC patients (9.8%) (*p* = 0.097).

### 3.3. Survival Outcomes

Our analysis revealed significant differences in survival outcomes between PHC and PBTC patients. The median OS for PHC patients was 12 months (95% CI: 10.4–13.6) compared to 9 months (95% CI: 7.6–10.4) for PBTC patients (*p* = 0.012) (Figure 1). Similarly, PFS differed significantly between the two groups, with PHC patients demonstrating a median PFS of 8 months (95% CI: 7.1–8.9) compared to 5 months (95% CI: 4.2–5.8) for PBTC patients (*p* = 0.008) (Figure 2). When stratified by the treatment regimen, PHC patients receiving mFOLFIRINOX demonstrated superior median OS (18.8 months, 95% CI: 16.5–21.1) compared to those receiving gem/nab-paclitaxel (9.7 months, 95% CI: 10.8–14.6) (*p* < 0.0001) (Figure 3). Similarly, PBTC patients treated with mFOLFIRINOX had longer median OS (14 months, 95% CI: 11.7–16.3) than those receiving gem/nab-paclitaxel (6 months, 95% CI: 4.7–7.3) (*p* = 0.011) (Figure 4). After adjusting for potential confounders, including age, gender, ECOG status, tumor stage, and CA 19-9 levels, in multivariate Cox proportional hazards models, tumor location remained an independent prognostic factor for both OS (adjusted HR for PBTC vs. PHC: 1.80, 95% CI: 1.53–2.11, *p* < 0.001) and PFS (adjusted HR: 2.39, 95% CI: 1.96–2.91, *p* < 0.001). Surgical resection was the strongest predictor of improved survival in both groups (adjusted HR: 3.99, 95% CI: 3.41–4.68, *p* < 0.001).

### 3.4. Latent Class Analysis

We performed latent class analysis (LCA) to identify distinct patient subgroups within our cohort of 400 PHC and 204 PBTC patients. The model incorporated demographic variables (age, gender, ethnicity), clinical parameters (ECOG status, disease stage, CA 19-9 levels), and treatment characteristics (chemotherapy regimen, treatment intent, surgical status, radiation therapy). Based on fit statistics and clinical interpretability, a three-class solution was optimal for both cancer types.

#### 3.4.1. Pancreatic Head Cancer Classes

In the PHC cohort (n = 400), three distinct classes emerged, shown below in Table 1:

Class 1: Advanced Disease—Palliative Class (76%, n = 304). This largest class consisted of 160 females and 144 males, predominantly Caucasian (n = 236, 77.6%). Most patients had Stage IV disease (n = 139, 45.7%) and received primarily palliative treatment (n = 169, 55.6%). First-line therapy was balanced between mFOLFIRINOX (n = 173, 56.9%) and gem/nab-paclitaxel (n = 131, 43.1%). Notably, this class contained 99 of the 103 adjuvant therapy recipients (32.6% of the class), but almost no patients received neoadjuvant therapy (n = 1, 0.3%) or radiation (n = 0). Median OS was 9 months, significantly shorter than the overall PHC median of 12 months reported in Section 3.3.

Class 2: Mixed Treatment—Intermediate Outcome Class (12%, n = 48). This class included 26 females and 22 males, with a balanced distribution of disease stages. The treatment approach was characterized by equal utilization of curative intent (n = 26, 54.2%) and palliative settings (n = 22, 45.8%), with limited neoadjuvant therapy (n = 1, 2.1%). Distinctively, this class had the highest proportion of radiation therapy utilization (n = 46, 95.8% of the class, representing 75.4% of all PHC radiation treatments). Median OS was 13.5 months, exceeding the overall PHC median of 12 months.

Class 3: Neoadjuvant—Surgical Class (12%, n = 48). Comprising 30 females and 18 males, this class was distinguished by the highest utilization of neoadjuvant therapy (n = 34, 70.8% of the class, representing 69.4% of all PHC neoadjuvant treatments) and substantial surgical intervention (n = 36, 75.0%, compared to the overall PHC surgical rate of 37%). Radiation therapy was utilized in one-third of patients (n = 16, 33.3%). Median OS was 18 months, substantially higher than the overall PHC median.

#### 3.4.2. Pancreatic Body/Tail Cancer Classes

For PBTC patients (n = 204), three classes were identified, shown below in Table 2:

Class 1: Extended Treatment—Better Outcome Class (36.8%, n = 75). This class included 28 females and 47 males, with a balanced distribution of disease stages, and was characterized by extended treatment duration (mean 10.4 cycles) and predominantly mFOLFIRINOX regimens (n = 51, 68.0% of the class). While palliative treatment was common (n = 48, 64.0%), this class contained the majority of PBTC adjuvant therapy recipients (n = 17, 22.7% of the class, representing 70.8% of all PBTC adjuvant treatments). Median OS was 16 months, substantially exceeding the overall PBTC median of 9 months.

Class 2: Advanced Disease—Limited Response Class (52.5%, n = 107). The largest PBTC class comprised 47 females and 60 males, with the highest concentration of Stage IV disease (n = 78, 72.9%). Treatment was characterized by shorter duration (mean 2.7 cycles), predominantly palliative intent (n = 90, 84.1%), and minimal adjuvant therapy (n = 7, 6.5%). Median OS was only 3 months, significantly below the overall PBTC median of 9 months.

Class 3: Radiation—Curative Intent Class (10.8%, n = 22). This smallest class (7 females, 15 males) was defined by exclusive use of radiation therapy (n = 20, 90.9% of the class, representing 100% of all PBTC radiation treatments) and the highest proportion of curative intent treatment (n = 11, 50.0%). This class also had the highest rate of surgical intervention (n = 14, 63.6%, compared to the overall PBTC surgical rate of 20.4%). Median OS was 9 months, matching the overall PBTC median.

#### 3.4.3. Clinical Implications

In both cancer types, classes with more intensive treatment approaches (neoadjuvant therapy, surgical intervention, extended treatment duration) demonstrated superior survival outcomes, consistent with our multivariate findings that surgical resection was the strongest predictor of improved survival (adjusted HR: 3.99). The composition of latent classes reflects the differences in treatment approaches between PHC and PBTC observed in Section 3.2. For example, the concentration of radiation therapy in specific classes (PHC Class 2 and PBTC Class 3) aligns with the overall radiation utilization rates of 15.3% and 9.8%, respectively. Across comparable treatment classes, PHC patients consistently demonstrated better survival outcomes than PBTC patients, supporting our multivariate finding that tumor location is an independent prognostic factor (adjusted HR for PBTC vs. PHC: 1.80 for OS). This latent class analysis provides additional granularity to our understanding of pancreatic cancer heterogeneity beyond anatomical location, potentially informing more personalized treatment strategies and prognostic assessment.

## 4. Discussion

PBTC usually carries a poor OS compared to PHC, probably due to the nature of the disease, which leads to delayed diagnosis and, thus, later treatment [24,36]. Molecular genomic alterations and genetic mutations also play a role in the OS among both PDAC types. Recent studies have identified molecular subtypes of PDAC with varying prognostic implications, with PBTC often harboring more aggressive genomic profiles compared to PHC [27]. Our study revealed consistent findings and showed superior OS rates among PHC patients compared to PBTC patients. Similar findings were reported in a real-world retrospective study, which demonstrated that PHC patients had better survival outcomes than PBTC patients, potentially due to earlier detection and differences in tumor biology [37,38].

A study by Zhang et al. aimed to explore the mechanism that molecular differences between PHC and PBTC lead to different clinical and survival outcomes using a next-generation sequencing (NGS) panel [39]. Their study revealed that PBTC carried a higher frequency of genomic alterations, particularly KRAS and SMAD4 mutations. SMAD4 was the predominant mutation in early stages (I–II) (56.0% vs. 26.5%, *p* = 0.021), while later stages (III–IV) showed significantly higher rates of KRAS mutations (100.0% vs. 75.8%, *p* = 0.001) compared to PHC at corresponding tumor stages [39]. Their study also revealed a higher frequency of MAPK pathway mutations in PBTC than PHC. The study concluded that PBTC was associated with more malignant outcomes at later tumor stages compared to PHC [39].

On the contrary, another study by Meng et al. used a matched cohort from the SEER database (2004–2014) to investigate the effect of tumor location in T1 resectable PDAC on the prognosis of PDAC. The study found that PHC had a worse prognosis in T1 resectable PDAC compared to PBTC. A multivariate analysis revealed that PBTC was associated with better chances of survival among T1 PDAC patients (hazard ratio HR, 0.69; 95% CI, 0.52–0.93; *p*  =  0.01). The study also revealed superior efficiency of the modified staging system in assessing survival rates compared to the AJCC 8th staging system. From this perspective, PHC appears to be more malignant compared to PBTC [37].

Another study by Micaily et al. from 2023 [21] used a large-scale, RWE database to retrospectively analyze the results of NGS of a cohort of 2015 patients. The study aimed to investigate the association of PDAC location with DNA damage response status (DRS) and its response to platinum-based therapy. It was found that PBTC had a higher frequency of DNA damage response pathway genomic alterations, such as BRCA1, BRCA2, and PALB2, at 21.7% of 618 cases compared to PHC, which had a frequency of 15.6% of 942 cases, with BRCA2 being a prominent contributor within this pathway (unadjusted *p*-value = 0.00244). In terms of treatment, PBTC exhibited a longer median PFS on first-line mFOLFIRINOX than gem/nab-paclitaxel regimens (*p* = 0.006393). No such difference was identified in PHC (*p* = 0.5546) [21]. These findings align with evidence that DDR pathway mutations, such as BRCA1/2, confer sensitivity to targeted therapies like olaparib in metastatic PDAC, particularly in patients with germline BRCA mutations [40].

To our knowledge, this is the first study to employ LCA and clustering analysis to identify subgroups of patients sharing similar demographic, clinical, and treatment-related characteristics and outcomes.

## 5. Conclusions

This study found that PHC is associated with superior survival outcomes relative to PBTC. Both tumor subtypes showed a better response to mFOLFIRINOX compared to gem/nab-paclitaxel, as supported by clustering analysis. Further studies are required to validate these findings and the results of the cluster analysis to provide a more personalized treatment approach and enhance patient outcomes in the treatment of PDAC.

## Figures and Tables

**Figure 1 cancers-17-01777-f001:**
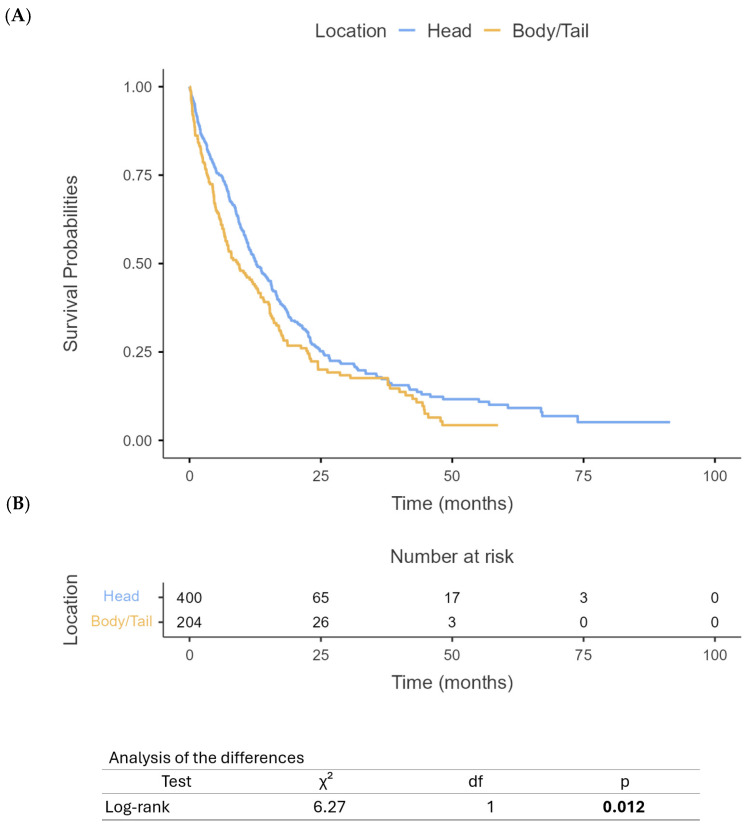
Overall survival based on PHC vs. PBTC (**A**) and number at risk table (**B**).

**Figure 2 cancers-17-01777-f002:**
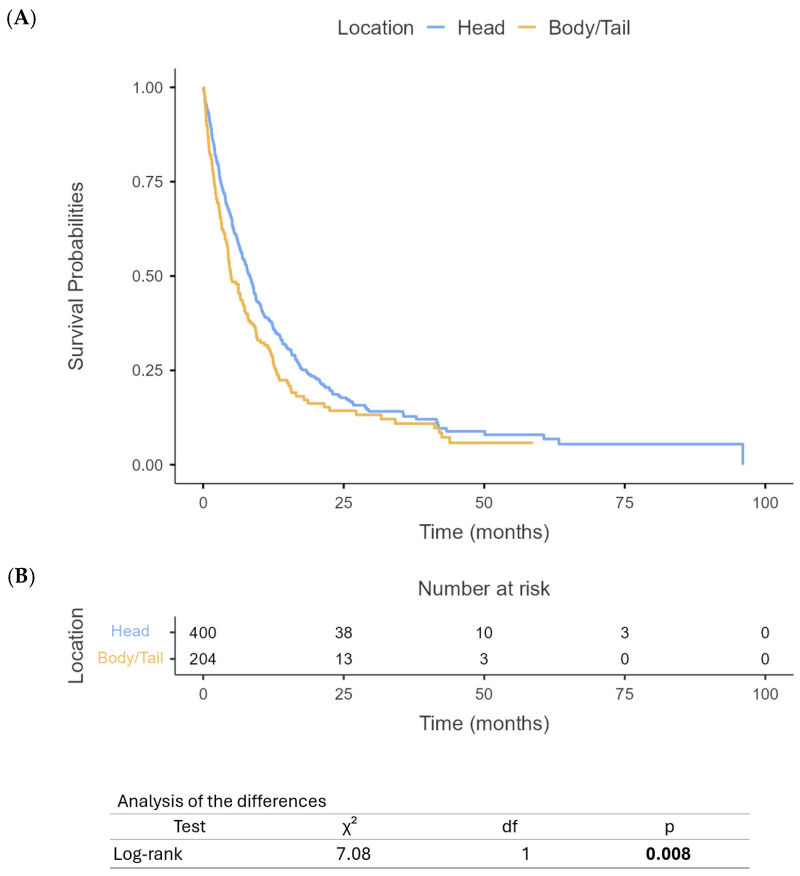
Progression-free survival based on PHC vs. PBTC (**A**) and number at risk table (**B**).

**Figure 3 cancers-17-01777-f003:**
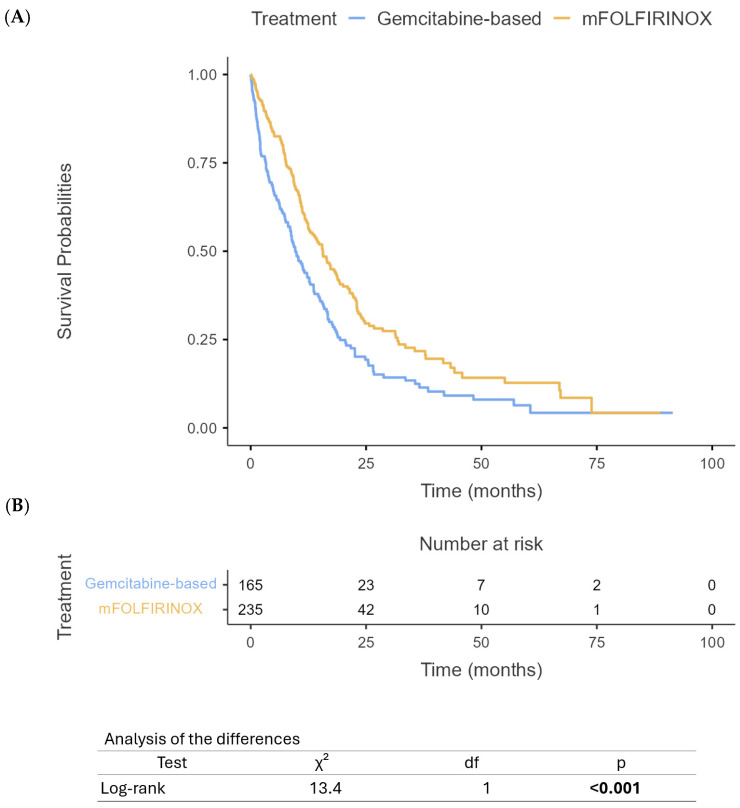
Overall survival of PHC based on treatment with mFOLFIRINOX or gem/nab-paclitaxel (**A**) and number at risk table (**B**).

**Figure 4 cancers-17-01777-f004:**
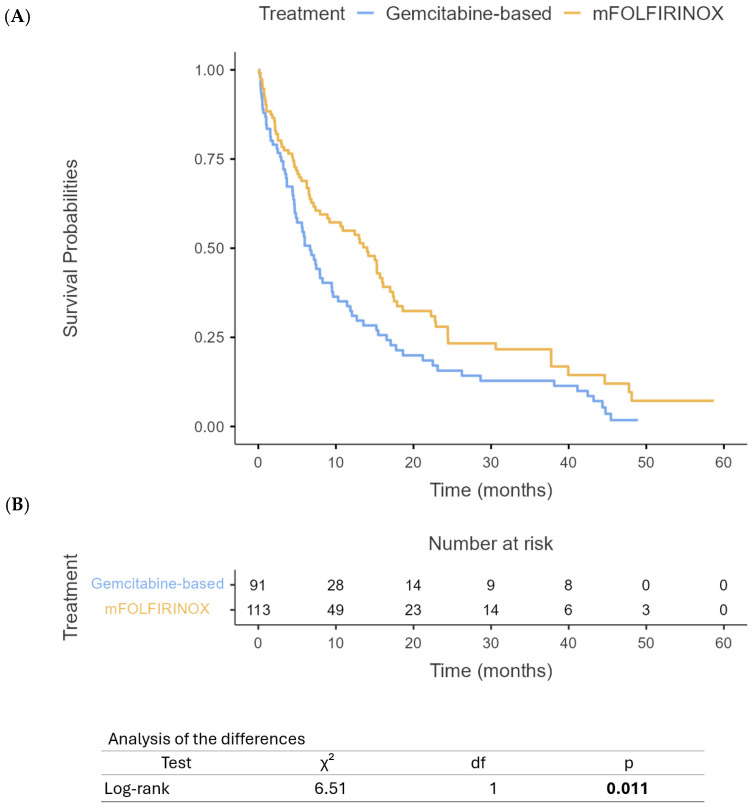
Overall survival of PBTC based on treatment with mFOLFIRINOX or gem/nab-paclitaxel (**A**) and number at risk table (**B**).

**Table 1 cancers-17-01777-t001:** Clustering analysis of PHC.

Category	Subcategory	C1	C2	C3
Gender Breakdown	Female	160	26	30
	Male	144	22	19
Ethnicity Breakdown	Caucasian	236	37	42
	Black	47	10	3
	Asian	14	0	3
	Hispanic	6	1	0
	Other	0	0	1
Treatment Breakdown	mFOLFIRINOX	173	27	37
	Gemcitabine based	131	21	12
Setting Breakdown	Palliative	169	18	8
	Adjuvant	99	3	1
	Curative	35	26	6
	Neoadjuvant	1	1	34
Number of Cycles	Mean	5.7	8.5	5.5
	Max	28	43	12
Pathologic Stage Breakdown	IV	139	3	1
	III	54	8	7
	IIB	53	10	11
	IB	18	10	14
	IIA	11	10	4
	IA	5	0	6
	II	3	4	2
	IIIA	1	0	0
	IIIB	2	0	0
Radiation	**Yes**	0	**46**	**16**
	No	304	2	33
Median OS	Median	9	**13.5**	**18**

**Table 2 cancers-17-01777-t002:** Clustering analysis of PBTC.

Category	Subcategory	C1	C2	C3
Gender Breakdown	Female	28	47	7
	Male	47	60	13
Ethnicity Breakdown	Caucasian	63	89	14
	Black	10	14	4
	Asian	2	2	2
	Hispanic	0	0	0
	Other	0	0	0
Treatment Breakdown	mFOLFIRINOX	51	46	14
	Gemcitabine based	24	61	6
Setting Breakdown	Palliative	48	90	9
	Adjuvant	17	7	0
	Curative	6	3	11
	Neoadjuvant	4	7	0
Number of Cycles	Mean	10.39	2.70	7.4
	Max	52	9	12
Pathologic Stage Breakdown	IV	46	78	3
	III	7	9	7
	IIB	7	7	1
	IB	7	2	6
	IIA	3	2	0
	IA	2	4	1
	II	0	0	0
	IIIA	0	0	0
	IIIB	0	0	0
Radiation	Yes	0	0	**20**
	No	**75**	107	0
Median OS	Median (months)	**16**	3	**9**

## Data Availability

The data of this retrospective analysis that support the findings of “Improved Survival in Pancreatic Head Cancer Compared to Pancreatic Body/Tail Cancer” are available upon request from the corresponding author, Maen Abdelrahim.

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
