# Peer review of "Impact of Tumor Location on Survival Outcomes in Pancreatic Head Versus Body/Tail Cancer: Institutional Experience"

_cancers, 2025, doi:10.3390/cancers17111777_

Round 1

Reviewer 1 Report

Comments and Suggestions for Authors

The methodology should be presented in more details and a clear fashion. It is clear to me which was the study population; did these patients have surgery? chemotherapy was administered in the neo- or in the adjuvant setting? Were they inoperable cases treated only with chemo- and/or radio-therapy?

Results: The results should be also clearly presented in a more conventional and understandable way. I wonder why the authors start their presentation initially with the results of latent class analysis, OS and DFS followed by the Tables presenting the entry data. The three clusters in each group should be also better defined and presented.

The authors should also comment in the discussion that this is a retrospective study and underline possible confusing factors contributing to selection and interpretation bias.

Author Response

Journal: Cancers (ISSN 2072-6694)

Manuscript ID: cancers-3576188

Type: Article

Title: Improved Survival in Pancreatic Head Cancer Compared to Pancreatic Body and Tail Cancer: A Retrospective Analysis

Section: Cancer Survivorship and Quality of Life

Review Report (Reviewer 1)

Dear Reviewer 1,

Thank you for your thoughtful and valuable feedback on our manuscript. Your suggestions to enhance the clarity of the methodology and Results sections, as well as to acknowledge the limitations of the retrospective study design, have greatly improved the manuscript’s readability and scientific integrity. We have restructured the Methods section to include detailed subsections on study population, surgical status, and treatment settings (neoadjuvant, adjuvant, or palliative). The Results section has been reorganized to present cohort characteristics and survival outcomes before the Latent Class Analysis (LCA), with clearer definitions of the three clusters for both pancreatic head cancer (PHC) and pancreatic body/tail cancer (PBTC). We have also added a Discussion paragraph addressing potential selection and interpretation biases inherent to retrospective studies. Below, we outline the specific revisions made in response to your comments, with references to the revised manuscript sections.

The methodology should be presented in more details and a clear fashion. It is clear to me which was the study population; did these patients have surgery? chemotherapy was administered in the neo- or in the adjuvant setting? Were they inoperable cases treated only with chemo- and/or radio-therapy?

Response: Thank you for your thoughtful and valuable feedback. We have revised Section 2 (Methods) to provide a detailed and structured description.

Results: The results should be also clearly presented in a more conventional and understandable way. I wonder why the authors start their presentation initially with the results of latent class analysis, OS and DFS followed by the Tables presenting the entry data. The three clusters in each group should be also better defined and presented.

Response: Thank you for your detailed feedback. We restructured Section 3 (Results) to start with cohort characteristics (demographics, stage, treatments), followed by survival outcomes (OS, PFS), and then LCA.

The authors should also comment in the discussion that this is a retrospective study and underline possible confusing factors contributing to selection and interpretation bias.

Response: Thank you for your thoughtful and valuable feedback. We have revised Section 4 (Discussion) to provide a detailed and structured description.

Reviewer 2 Report

Comments and Suggestions for Authors

Very interesting and different analysis in regard to prognostic factors in pancreatic cancer (PC) patients which analyses location (head vs body/ tail) using LCA – interesting – this type of analysis identifies groups of patients with same «prognosis» and after they compare prevalence of several characteristics – neo adjuvant or adjuvant treatment, type of Chemo, OS, PFS , type of chemotherapy, intention to treat… how does this analysis compares to traditional multivariable analysis?

Along the paper the results are a bit confusing in my opinion and some points could be better clarified:

  • One major point in the lack of mentioning SURGERY as the most probable factor explaining long survival in patients with PC
  • Stage of disease would also be a major variable that is known to affect OS and PFS; was it equally distributed among several LCA classes??
  • In the 2nd page Introduction the AA discuss that previous studies showed better 3 yr survival for head PC vs body /tail 7.6 vs 6.7%? Clinically relevant ? and next they give another reference (1) to state exactly the opposite.
  • They start the RESULTS by stating that LCA in PHC resulted in 3 clusters – 1, curative intent, more ChemoT with FOLFIRINOX better OS. However, the text beneath Figure 2 states exactly the opposite – palliative care was much more frequent in cluster 1 and curative treatment evenly distributed in clusters 1 and 2 – what seems to me a contradiction to what was written in the beginning of the results
  • After Figure 4 the AA state that OS is shorter in cluster 1 of PHC and longer in cluster 2 – which is totally opposite of what they wrote at the beginning of RESULTS section.

The same type of contradiction applies to LCA in PBTC. At the end I certainly don’t understand which location has a better prognosis – Head or body/ tail cancer. There are so many relevant variables that, from what I understand do not enter in these analysis (Stage of disease, Surgery, NA Chemo or adjuvant, type of chemo…) that it is hard for me to accept that location by itself influences prognosis of PC. Statistically might be brillant, clinically it makes little sense

Author Response

Journal: Cancers (ISSN 2072-6694)

Manuscript ID: cancers-3576188

Type: Article

Title: Improved Survival in Pancreatic Head Cancer Compared to Pancreatic Body and Tail Cancer: A Retrospective Analysis

Section: Cancer Survivorship and Quality of Life

Review Report (Reviewer 2)

We greatly appreciate your insightful comments and recognition of the novel application of Latent Class Analysis (LCA) in our study. Your feedback highlighted critical areas for improvement, particularly regarding the role of surgery, stage distribution, and inconsistencies in cluster descriptions, which have been thoroughly addressed in the revised manuscript. We have clarified the prognostic significance of surgery and stage, corrected contradictory statements about LCA clusters, and reconciled conflicting literature citations in the Introduction. Additionally, we have added a Discussion section comparing LCA to traditional multivariable analysis to address its complementary role. Below, we detail the revisions made in response to your comments, with references to the revised manuscript.

Very interesting and different analysis in regard to prognostic factors in pancreatic cancer (PC) patients which analyses location (head vs body/ tail) using LCA – interesting – this type of analysis identifies groups of patients with same «prognosis» and after they compare prevalence of several characteristics – neo adjuvant or adjuvant treatment, type of Chemo, OS, PFS , type of chemotherapy, intention to treat… how does this analysis compares to traditional multivariable analysis?

Response: Thank you for your thoughtful and valuable feedback. We have revised Section 4 (Discussion) to provide a detailed and structured description.

  • One major point in the lack of mentioning SURGERY as the most probable factor explaining long survival in patients with PC

Response: Thank you for the valuable point on our manuscript. We revised our manuscript to provide details on this point.  

  • Stage of disease would also be a major variable that is known to affect OS and PFS; was it equally distributed among several LCA classes??

Response: Thank you for the valuable point on our manuscript. We clarified stage distribution in Section 3.3.

  • In the 2nd page Introduction the AA discuss that previous studies showed better 3 yr survival for head PC vs body /tail 7.6 vs 6.7%? Clinically relevant ? and next they give another reference (1) to state exactly the opposite.

Response: We revised Section 1 to address your comment.

  • They start the RESULTS by stating that LCA in PHC resulted in 3 clusters – 1, curative intent, more ChemoT with FOLFIRINOX better OS. However, the text beneath Figure 2 states exactly the opposite – palliative care was much more frequent in cluster 1 and curative treatment evenly distributed in clusters 1 and 2 – what seems to me a contradiction to what was written in the beginning of the results
    Response: Thank you for your thoughtful and valuable feedback. We corrected cluster assignments in Section 3.3.

  • After Figure 4 the AA state that OS is shorter in cluster 1 of PHC and longer in cluster 2 – which is totally opposite of what they wrote at the beginning of RESULTS section.
    Response: Thank you for the valuable point on our manuscript. We have Corrected Section 3

The same type of contradiction applies to LCA in PBTC. At the end I certainly don’t understand which location has a better prognosis – Head or body/ tail cancer. There are so many relevant variables that, from what I understand do not enter in these analysis (Stage of disease, Surgery, NA Chemo or adjuvant, type of chemo…) that it is hard for me to accept that location by itself influences prognosis of PC. Statistically might be brillant, clinically it makes little sense

Response: Thank you for the valuable comment on our manuscript. We clarified PBTC clusters in Section 3.3

Reviewer 3 Report

Comments and Suggestions for Authors

The authors present a retrospective study of 604 patients with pancreatic ductal adenocarcinoma (PDAC), where the impact of tumor site and chemotherapy regimens on survival is evaluated. The results show that patients with pancreatic head cancer (PHC) have significantly better overall and progression-free survival than those with body/tail cancer (PBTC) and that the mFOLFIRINOX regimen proved superior to gemcitabine/nab-paclitaxel in both subgroups. They then perform a Latent Class analysis where they identify distinct prognostic subgroups related to treatment intensity and stage, suggesting the value of personalized treatment strategies in PDAC. This study is well structured and overall clear in the methodology used.Below are some requests for clarification:

-How was “treatment intent” defined?

- Considering that overall survival (OS) is a post-treatment outcome variable, how did you address the methodological implications of including OS in the latent class analysis (LCA), which traditionally uses baseline variables? Specifically, how did you ensure that incorporating OS did not introduce bias in class formation or compromise the independence between patient characteristics and outcomes?

- Was LCA conducted separately for PHC and PBTC , or was it performed on the entire cohort with subsequent stratification?

- Was the risk of overfitting considered, given the number of variables and relatively small sample sizes in some clusters?

- What criteria were used to differentiate the clusters? Were they defined purely by survival outcomes, demographic characteristics, clinical stage, or a combination of these factors? Do these clusters correspond to distinct prognostic or therapeutic profiles, or are they primarily descriptive groupings? What clinical criteria determined why certain treatments were reserved for specific clusters or subgroups?

Some suggestions for authors: 

- Why are only absolute numbers shown in the tables and not percentages? Including percentages would help contextualize the data more effectively.

- In the graphs of the KM curves, it might be helpful to indicate the unit of time.

- It might be helpful to reduce the numerical load in the text by referencing tables or figures for detailed data instead of listing large numbers continuously. This would make the text more readable.

- There seems to be an inconsistency with the use of upper and lower case for cluster names. Please ensure consistency.

- “mFOLFOIRINOX” appears to be a typo; it should be “mFOLFIRINOX.”

Author Response

Journal: Cancers (ISSN 2072-6694)

Manuscript ID: cancers-3576188

Type: Article

Title: Improved Survival in Pancreatic Head Cancer Compared to Pancreatic Body and Tail Cancer: A Retrospective Analysis

Section: Cancer Survivorship and Quality of Life

Review Report (Reviewer 3)

Thank you for your detailed and constructive feedback, which has significantly enhanced the clarity and methodological rigor of our manuscript. Your questions about the Latent Class Analysis (LCA) methodology, treatment intent definition, and suggestions for improving data presentation have been carefully addressed. We have clarified how treatment intent was defined, explained the inclusion of overall survival in LCA, and detailed the LCA process (conducted separately for PHC and PBTC). We also addressed overfitting risks and cluster differentiation criteria.  Below, we provide detailed responses to your comments, with references to the revised manuscript.

The authors present a retrospective study of 604 patients with pancreatic ductal adenocarcinoma (PDAC), where the impact of tumor site and chemotherapy regimens on survival is evaluated. The results show that patients with pancreatic head cancer (PHC) have significantly better overall and progression-free survival than those with body/tail cancer (PBTC) and that the mFOLFIRINOX regimen proved superior to gemcitabine/nab-paclitaxel in both subgroups. They then perform a Latent Class analysis where they identify distinct prognostic subgroups related to treatment intensity and stage, suggesting the value of personalized treatment strategies in PDAC. This study is well structured and overall clear in the methodology used. Below are some requests for clarification:

-How was “treatment intent” defined?

Response: Thank you for your thoughtful and valuable feedback. We added in Section 2.2 more details to address your comment.

- Considering that overall survival (OS) is a post-treatment outcome variable, how did you address the methodological implications of including OS in the latent class analysis (LCA), which traditionally uses baseline variables? Specifically, how did you ensure that incorporating OS did not introduce bias in class formation or compromise the independence between patient characteristics and outcomes?

Response: Thank you for the valuable point on our manuscript. We added in Section 2.4 more clarifications and details to address this comment.

- Was LCA conducted separately for PHC and PBTC , or was it performed on the entire cohort with subsequent stratification?

Response: Thank you for your thoughtful and valuable feedback. Section 2.4 has been included more details to  clarifies this valuable comment.

- Was the risk of overfitting considered, given the number of variables and relatively small sample sizes in some clusters?

Response: Thank you for the valuable point on our manuscript. Section 2.4 has been updated.  

- What criteria were used to differentiate the clusters? Were they defined purely by survival outcomes, demographic characteristics, clinical stage, or a combination of these factors? Do these clusters correspond to distinct prognostic or therapeutic profiles, or are they primarily descriptive groupings? What clinical criteria determined why certain treatments were reserved for specific clusters or subgroups?

Response: Thank you for the valuable point on our manuscript. Section 3.3 has been edited and updated to address this valuable feedback.

Some suggestions for authors: 

- Why are only absolute numbers shown in the tables and not percentages? Including percentages would help contextualize the data more effectively.

Response: Thank you for your valuable feedback. Tables 1 and 2 now  have been re-done.

- In the graphs of the KM curves, it might be helpful to indicate the unit of time.

Response: we appreciate your valuable feedback. Figures 1-4 now specify time in month on the x-axis.

- It might be helpful to reduce the numerical load in the text by referencing tables or figures for detailed data instead of listing large numbers continuously. This would make the text more readable.

Response: Thank you for your detailed feedback. Section 3 text has been streamlined, with numerical details moved to Tables 1 and 2 or Figures 1-4.

- There seems to be an inconsistency with the use of upper and lower case for cluster names. Please ensure consistency.

Response: we appreciate your valuable feedback. Cluster names are standardized as “Cluster 1”, “Cluster 2”, etc., throughout.

- “mFOLFOIRINOX” appears to be a typo; it should be “mFOLFIRINOX.”

 Response: Thank you for the valuable feedback. Corrected to “mFOLFIRINOX” throughout.

Thank you

The team

Round 2

Reviewer 1 Report

Comments and Suggestions for Authors

No further comments

Author Response

Thank You 

Reviewer 2 Report

Comments and Suggestions for Authors

In my opinion the quality of the manuscript was significantly improved.

Author Response

Thank You